# Optimising Outcomes and Surveillance Strategies of Rectal Neuroendocrine Neoplasms

**DOI:** 10.3390/cancers15102766

**Published:** 2023-05-15

**Authors:** Rajaventhan Srirajaskanthan, Dominique Clement, Sarah Brown, Mark R. Howard, John K. Ramage

**Affiliations:** 1Kings Health Partners, ENETS Centre of Excellence, Institute of Liver Studies, King’s College Hospital, London SE5 9RS, UK; 2Department of Gastroenterology, King’s College Hospital, London SE5 9RS, UK; 3Department of Histopathology, Kings Health Partners, ENETS Centre of Excellence, King’s College Hospital, London SE5 9RS, UK; 4Department of Gastroenterology, Hampshire Hospitals NHS Foundation Trust, Basingstoke RG24 9NA, Hampshire, UK

**Keywords:** rectal neuroendocrine tumors, MRI, endoscopy, surveillance, grade, staging

## Abstract

**Simple Summary:**

Rectal neuroendocrine neoplasms are increasing in incidence due in part to increased use of colonoscopy for colon cancer screening. These lesions can be difficult to characterise by endoscopists and, therefore, may be removed using an inappropriate endoscopic technique. Rectal neuroendocrine tumours should be fully staged prior to resection and this will help determine the best endoscopic approach for removal. Post resection, some of these tumours require ongoing surveillance. This article provides a detailed review of current evidence available to optimise assessment and removal of these lesions and approaches to surveillance for patients with rectal neuroendocrine tumours.

**Abstract:**

Rectal neuroendocrine neoplasms are increasing in incidence, in part due to increased endoscopic procedures being performed for bowel cancer screening. Whilst most of these lesions are low-grade well-differentiated neuroendocrine tumours, they can have a varied clinical behaviour. Frequently, these lesions are incorrectly characterised at endoscopy and, therefore, incompletely excised using standard polypectomy techniques. Furthermore, some cases are not fully staged prior to or post resection. In this article we discuss the endoscopic and surgical options available to improve the likelihood of achieving an R0 resection and the staging procedures that should be used in these NETs. We also review factors that may suggest a higher risk of nodal involvement or recurrence. This information may help determine whether endoscopic or surgical resection techniques should be considered. In cases of R1 resection we discuss the management options available and the long-term surveillance options and when these should be offered to patients.

## 1. Introduction

Rectal neuroendocrine neoplasms (NENs) have an increasing incidence due in part to the increased colonoscopic screening being undertaken to reduce the incidence of colorectal cancer [1,2]. Many countries have established screening programmes suggesting routine colonoscopy from the age of 45 or triaging patients for colonoscopy using stool tests such as the faecal immunochemical test (FIT) [3], which have resulted in several patients being identified with rectal NETs as incidental findings. The data from Public Health England (PHE) and the Surveillance, Epidemiology and End Results (SEER) Program in the USA suggest an incidence of 1–2 per 100,000 individuals per year [4,5]. There is a global variation in rectal neuroendocrine tumour (NET) incidence with much higher incidence reported in Asia [6]. 

Historically, rectal NETs were regarded as benign tumours and, therefore, inadequate endoscopic resection was frequently performed, and incomplete staging was undertaken [7]. Consequently, some patients present with recurrent or metastatic disease later. Even though the WHO classification of NETs has removed the benign/non-malignant classification, it still exists in the ICD-10-CM code for benign carcinoid tumour of the rectum. There are clear data that demonstrate rectal NETs have malignant potential and that nodal disease can occur even with small sub-centimetre NETs [8,9]. Therefore, they require careful staging and a multidisciplinary approach to management to ensure best long-term outcomes for patients. 

This article highlights the advances in endoscopic assessment to enable selection of the optimal endoscopic resection approach and endoscopic and histological characteristics that would be considered as adverse and leading to a higher risk of recurrence. Management options for patients with incomplete resections (R1/R2) will also be discussed. 

There has been a change in nomenclature of neuroendocrine neoplasms, separating morphologically well-differentiated neuroendocrine tumours, whilst poorly differentiated tumours are termed neuroendocrine carcinomas (NECs) [10]. Endoscopic resection should not be considered for the poorly differentiated NEC subtype since all these patients require an oncological resection [11]. This article is focussed on well-differentiated NETs; hence, the term rectal NETs is used throughout the article. 

## 2. Staging Investigations

To identify which treatment option to consider for patients with localised rectal NETs with no evidence of nodal spread, accurate staging is required [11]. The current European Neuroendocrine Tumor Society (ENETS) guidelines recommend an MRI pelvis, for the assessment of nodal involvement, in lesions >10 mm. This should be undertaken prior to resection where possible. Whilst the risk of nodal involvement in <10 mm lesions is reported as low, it is described as 2–25% in several studies [9,12,13,14]. Therefore, in our institution we undertake formal staging with MRI for all lesions ≥5 mm. If any concerning local pathological lymphadenopathy is noted on MRI, then functional imaging should be undertaken, with the preferred imaging modality being a somatostatin receptor PET (SSTR-PET). Figure 1 demonstrates nodal disease identified on an MRI scan and Ga-68 DOTATATE PET in a patient with rectal NET. Nodal involvement can be present in 20–35% of these cases with lesions >10 mm, therefore it is important to have MRI pelvis and SSTR-PET imaging.

If considering endoscopic resection, it may be important to ensure a complete (R0) resection is achieved. Rectal endoscopic ultrasound (EUS) should be undertaken prior to endoscopic resection of all lesions >5 mm to ensure no muscle layer involvement prior to resection. For lesions less than 5 mm, rectal EUS may not be able to visualise the lesion [15].

For all lesions up to 20 mm, endoscopic resection could be considered, however, it is imperative that all patients have close endoscopic assessment of the lesion to assess concerning/adverse endoscopic findings. Tumour shape can be suggestive of metastatic disease, as sessile lesions are often non-metastatic, whilst semi-pedunculated or ulcerating or fungating lesions have a higher risk of nodal or metastatic disease [16]. The surface pattern of the lesion can also be helpful to determine the likelihood of metastatic disease, as a normal mucosal appearance or yellowish tinge to mucosa often indicates low risk of metastatic disease. The presence of hyperaemia or surface ulceration is strongly linked to metastatic disease [16,17]. 

All patients with lesions >20 mm should be considered for surgical resection. The risk of nodal involvement is >50% in these patients and an oncological resection should be performed [18]. All patients with rectal NETs >20 mm should have staging with MRI pelvis, SSTR-PET imaging and CT chest, abdomen and pelvis (CAP). A significant percentage of large rectal NETs could be grade 3, in which case an FDG-PET should be performed in addition to SSTR-PET imaging. 

## 3. Endoscopic Therapy

There are several different endoscopic resection options. These include standard snare polypectomy, endoscopic mucosal resection (EMR), EMR-band ligation, EMR-cap fitted, EMR-underwater, EMR-ligation (EMR-L), endoscopic submucosal dissection (ESD) and ESD muscle ligation (ML) [17,19,20,21]. 

Standard polypectomy should be avoided since there is a high risk of an incomplete procedure and this leads to difficulty resecting the residual disease [11]. Describing in detail the different endoscopic techniques is beyond the scope of this article. However, a brief outline of the techniques would be helpful to understand the reasons for different histological outcomes. Standard polypectomy without injection of a lifting solution often means that the margins of the lesion are incompletely excised. Using endoscopic mucosal resection techniques, a submucosal lifting solution is injected below the lesion, enabling the cutting device to have a higher chance of clearing the deep and lateral margins. Using cap and band ligation techniques, the lesions are drawn upwards and away from the bowel wall, again enabling a better clearance of the margins. Endoscopic submucosal dissection is undertaken by using a submucosal injection to lift the lesion and then carefully dissecting underneath the lesion with a knife to remove the lesion, which enables the endoscopist to clearly delineate the margins of the lesion and hence increases the likelihood of achieving an R0 resection. The multicentre French series demonstrated a low R0 resection rate with snare polypectomy (18%). The ESGE suggests using advanced endoscopic resection techniques for removal of rectal NETs [22]. The success rates of different EMR techniques have been evaluated in several studies [20]. Whilst these are superior to standard polypectomy, they still have high rates of involvement in the deep margin resection, especially in lesions greater than 5 mm. However, a recent study comparing EMR-L vs. ESD in lesions less than 10 mm did not demonstrate a significant difference in R0 resection rates between the two techniques [19]. Therefore, for smaller lesions the choice of resection technique can be determined by the endoscopist’s experience and assessment of the lesion. However, randomised data are limited. There is a meta-analysis of different resection techniques and also a study demonstrating cap vs. ESD for <10 mm rNETs. 

Even using ESD, there is still difficulty achieving an R0 resection of the deep margin. A newer technique, ESD-ML, which involves dissecting the circular muscle layer, has been developed and this has been shown to offer a higher rate of R0 resection compared to ESD, without any increased risk of complications [20]. 

Another new technique enabling a full thickness mucosal resection using a full thickness resection device (FTRD) has been developed. There is increasing interest in using this method to achieve an R0 resection since the device enables deep resection to the serosal layer to be performed and then an over the scope clip ensures no perforation occurs [23,24]. A large German series where 501 FTRD procedures were performed in 31 centres, of which 40 cases were of rectal NETs, reported complete resection status in all 40 cases with no major adverse events reported [24]. The median lesion size was 8 mm and procedure time was a short 18.5 min. This technique appears promising, especially in areas of the world where ESD is not available. The commonest complication was bleeding (10% of NET patients had minor bleeding post resection). In a multicentre study of 229 patients undergoing resection of different lesions by 22 endoscopists, the authors reported a moderate or severe complication rate of 17.5% [25]. The immediate bleeding rate was 5.7%, perforation 0.4% and delayed bleeding 6%. Its role in the treatment paradigm of rectal NETs is not fully elucidated and, where ESD can be performed by experienced endoscopists, this remains the preferred option. If there is no access to ESD, then FTRD may provide a better option at achieving an R0 resection than EMR but there are no randomised data to support this approach. 

If patients have had previous biopsies demonstrating grade 2 histology, the preferred approach would be ESD-ML, ESD or TEMS/TAMIS. The higher risk of recurrence and adverse prognosis mean a technique offering the best chance of achieving an R0 resection should be chosen. If patients have a high grade 2 histology, for example, a Ki67 10% with evidence of possible LV invasion, then a discussion about the approach to surgical resection, such as TEMS, should be undertaken or even an oncological resection could be considered. Patients with grade 3 rectal NET should discuss oncological resection to enable lymph node clearance. 

## 4. Histological Assessment Post Resection

The histological definition of tumour resections, either complete (R0) or incomplete (microscopic (R1) or macroscopic (R2)), is important to make sure data are comparable across different studied series. Technically, R0 indicates no tumour cells at the resection margin, whereas some authors have suggested no cells <1 mm from the resection line. The presence of cells at the margin of the resected specimen (R1) of a diathermied lesion does not necessarily imply NET cells at the cut surface that remains, since the diathermy may account for a small area of tissue that can be difficult to examine histologically.

## 5. Risk Factors for an R1 Resection

Several studies have examined the risks for an incomplete endoscopic resection. An R1 resection is classed as when there is tumour present at the resection margin. When using ESD techniques the lateral margins are often straightforward to delineate, however, it can be difficult to achieve an R0 resection at the deep margin due to the proximity to the muscle layer. 

The most common predictors of an R1 resection are tumour grade and size, as summarised in Table 1. Grade 2 and 3 tumours have a higher rate of incomplete resection which appears to be independent of the size of the lesion [26]. 

Studies have often reported size being a factor leading to R1 resection but this is not universally reported. Some studies have reported the presence of lymphovascular invasion and depth of invasion as being risk factors [28]. Additionally, some studies on univariate analysis have noted age to be a factor and location of the primary tumour within the rectum. A higher rate of incomplete resection was reported for lesions nearer the dentate line [27]. The significance of an R1 resection is unclear. There are a number of studies which have demonstrated no evidence of recurrence following incomplete resection of rectal NETs [27,30,32]. A study of 54 patients with R1 resections did not demonstrate any adverse outcomes, however, the duration of follow up was short and the method of surveillance was endoscopic only and, therefore, development of nodal or metastatic disease may not have been captured [27]. In this series all the lesions were <10 mm and over 90% were grade 1. The concerns arising with an R1 resection are the difficulty knowing if patients can be discharged from follow up safely and duration of follow up. Approaches to follow up are discussed later in this article but the importance of an R0 resection for G1 NETs under 10 mm is that they can be safely discharged if there are no adverse histological features. 

If the initial resection was an R2 resection, then a second endoscopic/surgical procedure should be performed to remove the residual tumour. This is often the case as demonstrated in the French multicentre series, in which 100 of 345 cases required a second procedure to remove residual disease [31]. 

## 6. Risk Factors for Recurrence

There is a paucity of data to help determine the risk of local or distant recurrence following resection. Therefore, some of the recommendations or risk factors (summarised in Table 2) are linked to those associated with the risk of developing metastatic disease at diagnosis or post resection. Following resection, it is important to characterise the following: firstly, whether there was macroscopically visible disease post resection (R2 resection) and, secondly, to determine whether this was an R0 resection. The latter will assess both the circumferential margins and the deep margin, as commonly it is the deep margin that is involved. This is because the tumours often arise below the mucosa and, like submucosal tumours, it can be difficult to achieve a clear deep margin. 

From published literature, the following factors are also likely to increase risk of recurrence: age and grade of tumour, with grade 2 or 3 conferring an increased risk. Interestingly, R1 resection is not an independent risk factor for recurrence [27,30,32]. The presence of lymphovascular invasion is thought to be a strong risk factor for metastasis formation [28]. 

For grade 2 or 3 histology the odds ratio for recurrence is significantly higher and hence it is important to achieve an R0 [28]. Lesion size is also associated with risk of nodal involvement and recurrence with lesions >10 mm having a higher risk of recurrence [27,33]. Histological features such as the presence of lymphovascular invasion increase risk of recurrence [33]. For grade 1 rectal NETs <10 mm in size there is a large amount of data and the risk of recurrence is low, <3%, at 5 years [34,35,36].

A study by Fahy et al. noted recurrence in 7 patients of 69 in the study population, and the time to recurrence was reported as 10–92 months, with a mean of 43.3 months. All were grade 2 disease and five of seven had primary tumours greater than 20 mm [37]. A study from Kojima et al., in 2019, reported recurrence in 3 cases of a study cohort of 79 occurring around 5–7 years post resection, and all cases were grade 2 disease, however, size of the lesion varied from 10–55 mm [26].

Risk factors which are more likely to lead to recurrence would be grade 2 or 3 NETs, presence of lymphatic or vascular invasion, size of lesion >10 mm and depth of invasion [26,28,38,39]. 

## 7. Approach to Incomplete Resection following Incidental Finding of a Rectal NET

Frequently, rectal NETs are resected prior to formal characterisation of the lesion. The endoscopists may incorrectly identify the lesion as an adenoma, atypical hyperplastic polyp or possible rectal lipoma. A multicentre study demonstrated only 18% of French endoscopists suspected a rectal NET at the time of the index endoscopic procedure [31]. By not identifying the lesion as a rectal neuroendocrine tumour at the initial assessment, and commonly misdiagnosing it, endoscopists will often biopsy the lesion or undertake a conventional snare polypectomy. Figure 2 demonstrates the endoscopic appearance of <10 mm rectal NETs and, using careful imaging and assessment, the difference of these lesions from a polyp or lipoma can be seen. Once the histology is available, along with any additional information in the report outlining the size of the tumour prior to intervention, the team then must determine the management. The approach should follow the recommendations in terms of appropriate staging imaging as for de novo lesions with repeat endoscopic evaluation by flexible sigmoidoscopy +/− EUS. This will determine the extent of residual disease and aid decision making in relation to the requirement for further endoscopic resection. Biopsy of the scar may also be helpful in addition to assessment by colonic ultrasound, especially if there is no evidence of recurrence on inspection of the site and on colonic ultrasound. A study by Steir et al. demonstrated that 46% of biopsies of a normal colonic ultrasound and scar at endoscopy demonstrated residual disease [40].

If the MRI scan does not demonstrate any evidence of nodal disease but there is residual macroscopic visible disease or disease noted on colonic ultrasound (Figure 3), discussion about a repeat procedure to remove residual disease should be undertaken with the patient. To achieve resection in this setting, ESD is often required to enable the best assessment of resection and achieve an R0 resection (Figure 3). There are data suggesting success in salvage ESD for cases with residual tumour post initial resection with low rates of recurrent disease [41]. There is also emerging evidence for the role of EndoRotor to enable the clearance of residual disease, though no data are published on NETs to date. Salvage ESD to resect the scar following R1 resection is recommended in the ESGE guidelines [22].

For grade 1 rectal NETs <10 mm without lymphovascular invasion who undergo R1 resection assessment of the scar with biopsy is reasonable to ensure no residual disease remains as these patients are at a low risk of recurrence. For patient with grade 2 or 3 rectal NETs with no residual disease at endoscopic assessment a discussion about ESD resection of the scar could be considered. For patients with visible residual disease at the surveillance assessment resection of the residual tumour and scar should be undertaken. The rationale being that if an R0 resection can be achieved, and the lesion fully staged, the need for surveillance can be determined. For lesions 10–20 mm with grade 1 histology and R1 resection margin, endoscopic assessment and MRI is required, and then resection of the scar or possibly full thickness resection should be considered. A proposed algorithm for management of incompletely excised rectal NETs has been described in Figure 4. 

## 8. Follow Up Recommendations

The evidence for follow up is based on experience and risks of recurrence. In our practice, we would recommend the following approaches (summarised in Table 3). 

If baseline imaging demonstrates no evidence of nodal or metastatic disease, then, for small lesions <10 mm without lymphovascular invasion and with a curative R0 resection, patients can be discharged with no further follow up required. 

For patients with rectal NET G1 <10 mm, however, with the presence of lymphovascular invasion, MRI and sigmoidoscopy follow up is required for 5 years. 

For patients with an R1 resection of <10 mm NET without further resection, then 12 monthly MRI and sigmoidoscopy could be considered. 

With grade 2 tumours, independent of size, and an R0 resection, follow up should be considered and this would be a combination of MRI and/or endoscopic follow up for 5 years. 

For patients with a grade 2 tumour <20 mm and R1 resection, then consideration of a repeat procedure for resection of the scar should be undertaken. 

A >10 mm lesion undergoing R0 resection (after either initial or subsequent resection) should be followed up for at least 5 years—6 monthly MRI and yearly sigmoidoscopy independent of the grade of tumour.

If >20 mm, these patients should have oncological resection and then follow up as per oncological protocols, with 6 monthly scans for 5 years. 

## 9. Conclusions and Future Direction

In summary, rectal NETs are often incidental findings at colonoscopy. They require careful assessment and staging prior to resection. Tumour size and grade are predictive factors for the risk of nodal metastases, therefore, appropriate staging investigations should be undertaken [42]. Choice of the type of resection depends in part on the size of the lesion, local expertise and findings from the staging investigations. There are limited data on the most effective endoscopic resection technique with comparable data using ESD vs. EMR-band ligation. However, standard polypectomy or cold snare polypectomy is unlikely to yield a clear resection and should be avoided. Education in lesion recognition is required for endoscopists to aid assessment and this may be augmented with the advent of artificial intelligence. Post resection, patients may require long-term follow up depending on the histological findings. There remain many unmet needs in patients with rectal NENs, in terms of optimal resection strategy and when to discharge patients. Large cohort studies to better determine prognostic factors and understand the global geographic variation in rectal NETs would be helpful. 

## Figures and Tables

**Figure 1 cancers-15-02766-f001:**
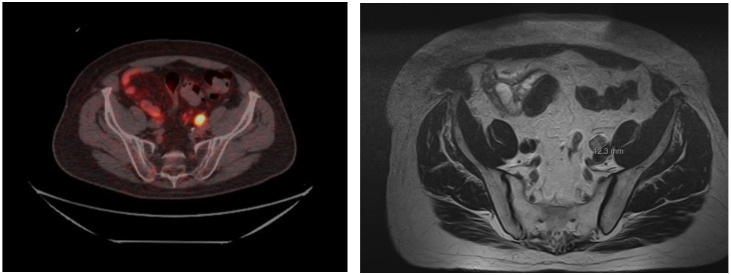
**Left** image illustrates a Ga-68 DOTATATE PET-avid lymph node in a patient with 1 cm rectal NET. **Right** image is a staging MRI scan for this patient which demonstrates a 12 mm pathological lymph node.

**Figure 2 cancers-15-02766-f002:**
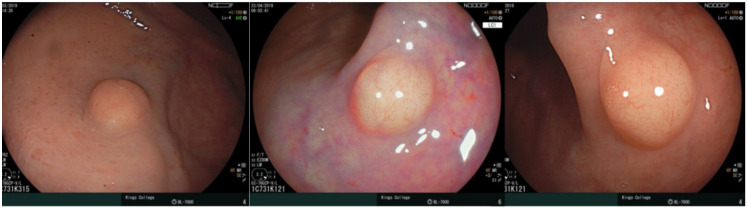
Endoscopic imaging of small rectal NETs. Figure on the left demonstrates the mucosal appearance of the lesion. Middle image highlights the yellow hue seen when using linked colour imaging and helps differentiate the lesion from hyperplastic polyps. The right panel is rectal NET demonstrating the submucosal appearance which differentiates it from mucosal polyps such as hyperplastic polyps.

**Figure 3 cancers-15-02766-f003:**
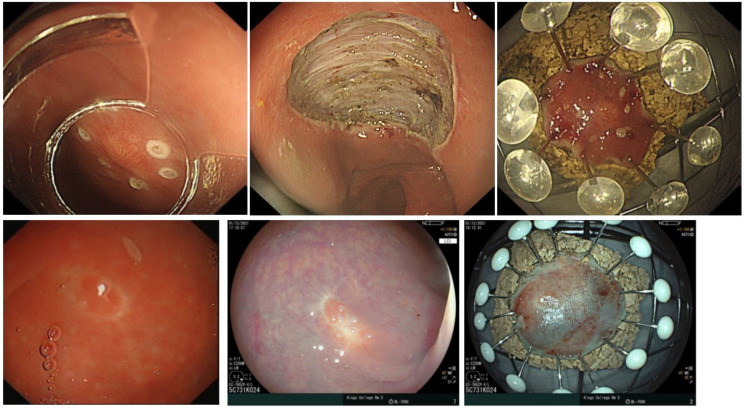
Top three panels: recurrence of a previously resected rectal NET as noted by the submucosal protrusion and confirmed with colonic US findings. This area was resected with ESD. Top middle panel notes the clean base and sample is pinned on cork board to aid pathological assessment. The lower panel notes residual NET present, which is better assessed with linked colour imaging, the residual tumour demonstrating a yellow hue. The resected sample can be seen in the lower right panel.

**Figure 4 cancers-15-02766-f004:**
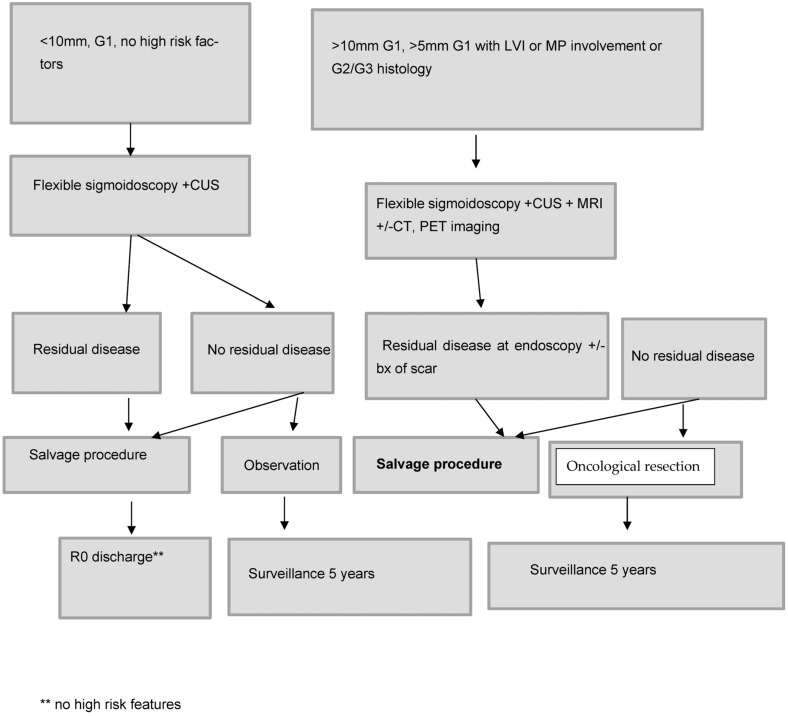
Proposed algorithm for approach to R1 resection of rectal NETs.

**Table 1 cancers-15-02766-t001:** Summary risk factors for R1 resection.

Risk Factor	Odds Ratio and 95% CI	*p*-Value	Reference
Grading of tumour (G2–G3)	Kojima no ORR 3.59 (1.37–9.42) 2.36 (1.14–4.89)	<0.01 0.01 0.02	Kojima, 2019 [26] Wang, 2021 [15] Li, 2021 [27]
Lymphovascular invasion	No ORR provided		Nam, 2022 [28]
Depth of invasion	No ORR provided		Nam, 2022 [28]
Nearer dentate line	No ORR provided		Li, 2021 [27]
Size of lesion >5 mm	2.393 (0.70–8.09)	0.16	Choi et al., 2017 [29]
Central depression on surface, hyperaemic pattern	11.529 (2.37–55.92) 7.49 (2.26–24.76)	0.002 0.001	Choi et al., 2017 [29] Zhuang et al., 2021 [30]
Type of endoscopic resection	10.8 (4.3–31.5)	<0.001	Fine et al., 2021 [31]

**Table 2 cancers-15-02766-t002:** Summary of risk factor recurrence.

Risk Factor	Odds Ratio and 95% CI	*p*-Value	Reference
Age	Multivariate analysis 0.97 (0.91–1.03)	0.263	Nam, 2022 [28]
Grading G2–G3	Multivariate (Wang) 10.03 (2.21–45.32) Multivariate (Nam) 6.28 (1.21–32.53)	0.002 0.03	Wang, 2021 [15] Nam, 2022 [28]
Lymphovascular invasion	No odds ratio reported		Chida, 2019 [33]
Size > 20 mm	Multivariate analysis 14.66 (1.08–198.58)	0.04	Wang, 2021 and Chida, 2019 [15,33]
Depth of invasion	Multivariate analysis 13.64 (1.78–104.66)	0.01	Wang, 2021 and Chida, 2019 [15,33]

**Table 3 cancers-15-02766-t003:** Summary of follow up advice that we give at our institution.

Grade	Size	LVN	Type of R Resection	FU Advice
G1	<10 mm	Negative	R0	Discharge
	<10 mm	Positive	Any R resection	MRI and flexible sigmoidoscopy every 12 months for 5 years
	<10 mm		R1 resection	Consider MRI and flexible sigmoidoscopy
G2	<20 mm	Any	R0 resection	MRI and flexible sigmoidoscopy for 5 years
	<20 mm	Any	R1 resection	Repeat resection, followed by 6 monthly MRI and annual flexible sigmoidoscopy
	>20 mm	Any	Any R resection	Oncological resection followed by 6 monthly MRI for 5 years

## Data Availability

The data can be shared up on request.

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
