# Peer review of "Optimising Outcomes and Surveillance Strategies of Rectal Neuroendocrine Neoplasms"

_cancers, 2023, doi:10.3390/cancers15102766_

Round 1
Reviewer 1 Report
Srirajaskanthan et al. provided a manuscript that reviews the current evidence to assess and manage rectal neuroendocrine tumors.
The authors provided a good introduction citing the increased discovery of rectal NETs. The staging tools were discussed by tumor size with recommended intervention based on the risk of nodal metastasis. Several endoscopic techniques to resect rectal NETs were discussed, including a full-thickness resection device. The risks of tumor recurrence were adequately discussed.
Overall, the manuscript was succinctly written with a focus mainly on the endoscopic management of rectal NETs. It was easy to read and well-structured. I have a couple
1. Lines 138-141, please expand the treatment option of G2 NET that surgical resection should be considered given the risk of nodal metastasis and recurrence.
2. Please create a flow chart by tumor size to include the intervention options in steps, followed by a surveillance plan to summarize the manuscript.
Author Response
Thank you very much for your interest in our article and for the constructive comments your reviewers kindly made. We have answered the reviewer’s comments in full and made the appropriate changes as outlined. Below is the response to each individual point.
Reviewer: 1
- Lines 138-141, please expand the treatment options of G2 NET that surgical resection should be considered given the risk of nodal metastases and recurrence.
We agree with the reviewer that oncological surgical resection should sometimes be considered in these cases. I have included two new sentences addressing this point (section 3, last paragraph)
- Please create a flow chart by tumour size to include the intervention options in steps, followed by a surveillance plan to summarize the manuscript.
We have created a treatment algorithm based on size to determine the choice of intervention, see new Figure 4. It would be too complicated to add surveillance to this figure and this remains well summarized in table 3.
Reviewer 2 Report
This manuscript provides a review of existing literature on the clinical management of rectal neuroendocrine tumors, including aspects such as clinical staging, what factors are considered to increase a patient’s risk of incomplete resection and recurrence, and differences in endoscopic treatment strategies. The authors did a great job bringing together research on a topic that remains relatively understudied. However, there are key issues that should be addressed prior to acceptance for publication:
Major suggestions
1. The organization of the text should be improved, since currently there is a lack of flow that has the reader jumping between sections and re-reading similar sentences in different sections. In this sense, the manuscript requires additional effort to become comprehensive and concise. Several sentences throughout the text also require careful proof reading. The manuscript would benefit from being reviewed by a language editor.
2. The manuscript can also be improved by expanding the critical assessment of cited work, and discussing some of the implications and the limitations of those studies in more detail.
3. The tables should be expanded and include additional details, such as additional factors that have been historically considered to be associated with risk of recurrence or R1 resection. I recommend showing negative studies too (what was not associated with risk of R1 resection? Did an EMR vs polypectomy reduce the odds of locoregional recurrence?). This will make the tables more useful for the reader.
Minor comments
- The section about endoscopic therapy needs further elaboration of each technique. For example, EMR-capfitted, EMR-underwater, EMR-l are not discussed. I recommend adding indications/contraindications and approaches to technical considerations when picking one therapeutic method over the other.
- Line 106-107: please present some data behind the success rates. You say several studies, but only cite one. Also, elaborate on what metrics to which EMR is superior over standard polypectomy.
- Line 111: please present the data on R0 resection between the techniques.
- German trial results: please provide the frequencies of adverse events (lines 129-130).
- Lines 129-133: This multicenter study needs clarity. What is the purpose of this citation? There are multiple important points you can use this citation for – what was the success rate of using FTRD? What was the size of the tumors? The rate of R- resection? Critically compare this to the German series: why was the R0 resection rate lower in this study compared to the German series?
- Risk factors for R1 resection:
o Table 1: There are a number of factors which have been identified as increasing the risk of an R1 resection within the rectum. This includes the mode of endoscopic resection (as stated previously in the manuscript), gender and lesion size. I recommend preparing a more comprehensive table which also reviews factors which have not been associated with (or reduced) the odds of having an R1 resection.
o Lines 164-167: this is an excellent point but needs careful elaboration. Please present the cohort size, tumor characteristics (and tumor biology), the methodology and rates of outcomes in this paper. In this study, a number of patients were lost to follow up, a small number underwent surveillance, and many required additional procedures. Not only was the follow-up too short, sample size was small. Additionally, these patients might have had good tumor biology which resulted in a low frequency of locoregional or distant recurrence.
o Please elaborate on the French multicenter series (lines 174-175).
- Risk factors for recurrence:
o Lines 184-185: please clarify what you mean by “behave”. How does it make a difference? Is the behavior a change from a slow growing tumor to a more aggressive one? Does it impact survival?
o Lines 189-190: please clarify what citation was used for the statement “R1 resection is not an independent risk factor for recurrence”. This should be in table 2, and it needs a critical review. Is this because you don’t have long enough FU in these studies? Or because you are presenting on well-differentiated NETs?
o As stated previously, Table 2 should also be more comprehensive.
- Approach to incomplete resection following incidental finding of a rectal NET
o Line 229: please elaborate on the data which presents success in salvage ESDs in patients with residual tumor. This needs a critical review.
- SEER database should be described as SEER program, and the first letter of each should be in upper-case: “Surveillance, Epidemiology, and End Results...”
- Line 44: please clarify, “neuroendocrine tumor (NET)
- Line 67: please identify what ENETS guidelines are, as you did with SEER and NEN.
- Line 124: German trial requires citation.
- Line 189: please fix “1. Age, 2. Grade 2 or 3 histology”. This is not an appropriate way to present in a manuscript.
- Line 197 and 198: is this locoregional or distant recurrence? Or both?
Author Response
Reviewer 2
Major suggestions
- The organization of the text should be improved, since currently there is a lack of flow that has the reader jumping between sections…
We have significantly revised the text and organisation of the document and this is throughout the manuscript. All authors have repeatedly re-read the revised manuscript.
- The manuscript can also be improved by expanding the critical assessment of cited work, discussing some of the implications and the limitations of those studies in more detail.
We have expanded the cited literature, including more studies as suggested in areas of comparison of endoscopic techniques. Recent relevant articles published in the last 4 months have also been incorporated to the review. The limitations of some of the studies cited has been highlighted throughout the study (see section of endoscopic resection technique, FTRD etc..)
- The tables should be expanded to include additional details
Table 1 has been expanded to include other risk factors associated with an R1 resection (see, table 1).
Minor comments
- The section about endoscopic therapy needs further elaboration of each technique
This has been expanded (section 3, second paragraph)
- Line 106-107: please present some data behind the success rates. Also, elaborate on what metrics to which EMR is superior over standard polypectomy.
Further studies have been cited to demonstrate success rates of achieving an R0 resection using different endoscopic resection techniques. The main data source for the role of standard polypectomy vs. EMR is included from the excellent French multicentre study (section 3, second paragraph)
- Line 111: please present the data on R0 resection between different techniques .
This has been incorporated in to the text
- German trial results: please provide the frequencies of adverse events
This has been incorporated in to the text, section 3, paragraph 4.
- Lines 129-133. This multicentre study needs clarity.
Agreed, more clarity has been given regarding the nature of complications , (section 3 paragraph 4).
Risk factors for R1 resection
- Table 1 There are a number of factors which have been identified as increasing the risk of an R1 resection in the rectum.
Table 1 has been significantly revised and includes a more comprehensive list of risk factors for an R1 resection.
- Lines 164-167: this is an excellent point but needs care elaboration. Please present the cohort size, tumour characteristics (and tumour biology), the methodology and rates of outcomes in this paper.
This section has been expanded to reference new articles and expand on the study cohort.
- Lines 184-185: please clarify what you mean by ‘behave’. How does it make a difference? IS the behaviour a change from a slow growing tumour to a more aggressive one? Does it impact survival?
The wording has been changed since ‘behave’ is the wrong term.
- Line 189-190: R1 resection is not an independent risk factor for recurrence
This has been referenced by citing the published literature to date.
- As stated previously, Table 2 should also be more comprehensive
Approach to incomplete resection following incidental finding of rectal NET
- Line 229: please elaborate on the data which presents success in salvage ESDs in patients with residual tumour. This needs critical review.
This section has been amended to include more recent data.
- SEER – this has been changed
- Line 44 – NET this has been changed
- Line 67: ENETS – this has been changed
- Line 124 :German trial needs citation – this has been cited.
- Line 189 – this has been changed
- Line 197 and 198: is this locoregional or distant recurrence?
This is for both locoregional and metastatic disease